# Etiology of Acute Leukemia: A Review

**DOI:** 10.3390/cancers13092256

**Published:** 2021-05-08

**Authors:** Cameron K. Tebbi

**Affiliations:** Children’s Cancer Research Group Laboratory, 13719 North Nebraska Avenue, Tampa, FL 33613, USA; ctebbi@childrenscancerresearchgrouplaboratory.org

**Keywords:** etiology, leukemia, acute lymphoblastic leukemia, acute myeloblastic leukemia, genetics, causes, occupations, hobbies, genetic, infections, mycovirus, *Aspergillus*

## Abstract

**Simple Summary:**

Acute leukemias are some of the most common cancers affecting all age groups. Despite a significant improvement made in the treatment of acute leukemias, their cause remains unknown. A number of genetic and environmental factors for the development of acute leukemias have been proposed, but none have been proven. Undoubtedly, genetics have a major role in the development of these diseases. The effects of a variety of environmental factors, occupations and hobbies have been explored. A recent “two-hit” theory” for the development of acute lymphoblastic leukemia has been proposed. This combines genetic factors and exposure to infections for the development of this disease. Several genetic factors are suggested. Most recently, for the infection portion, exposure to a virus containing *Aspergillus Flavus* has been proposed. This review summarizes what is currently known about the factors that are proposed for the development of acute leukemias.

**Abstract:**

Acute leukemias constitute some of the most common malignant disorders. Despite significant progress made in the treatment of these disorders, their etiology remains unknown. A large and diverse group of genetic and environmental variables have been proposed. The role of a variety of factors, including pre-existing and acquired genetic mutations, exposure to radiation and various chemicals during preconception, pregnancy and throughout life, have been explored. The effects of inherited genetic variations and disorders, pre-existing diseases, infectious agents, hobbies, occupations, prior treatments, and a host of other factors have been proposed, but none is universally applicable to all cases. Variation in the incidence and prognosis based on the age, sex, race, type of the disease, geographic area of residence and other factors are intriguing but remain unexplained. Advances in genomic profiling, including genome-wide gene expression, DNA copy number and single nucleotide polymorphism (SNP) genotype, may shed some light on the role of genetics in these disparities. Separate two-hit hypotheses for the development of acute myeloblastic and lymphoblastic leukemia have been proposed. The latter combines genetics and infection factors resulting in leukemogenesis. A number of pre- and post-natal environmental conditions and exposure to infections, including a mycovirus infected *Aspergillus flavus*, have been suggested. The exact nature, timing, sequence of the events and mechanisms resulting in the occurrence of leukemia requires further investigations. This review summarizes some of the above factors in acute lymphoblastic and myeloblastic leukemias and the direction for future research on the etiology of these disorders.

## 1. Background

Leukemia is one of the most common malignant disorders affecting the world population. Globally, in 2018, leukemia ranked as the fifteenth most common diagnosed cancer with 437,033 cases and 309,006 mortalities, amounting to the eleventh cause of death due to malignant disorders [1]. The geographic distribution of leukemia is universal, with higher prevalence and overall mortality in the more developed countries. The mortality rate, however, is higher in developing countries. A detailed pattern of the incidence of cancer in general, and leukemia in particular, is available [1]. Based on the Cancer Facts and Figures provided by the American Cancer Society, for the year 2020, it was estimated that 178,520 individuals were to be diagnosed with leukemia, lymphoma and myeloma in the United States. This accounts for 9.9 percent of the estimated 1,806,590 new cancer cases diagnosed in that year. While both sexes are affected, leukemia is more prevalent in males. The age-standardized incidence rates for leukemia in males and females in 2018 in the United States were 6.1 and 4.3 per 100,000, respectively. Likewise, the mortality rate of 4.2 per 100,000 population was higher for males compared to 2.8 per 100,000 in females [1].

Acute leukemias are malignant clonal disorders of blood-forming organs involving one or more cell-lines in the hematopoietic system. These disorders are marked by the diffuse replacement of bone marrow with abnormal immature and undifferentiated hematopoietic cells, resulting in reduced numbers of erythrocytes and platelets in the peripheral blood. Based on the origin of the abnormal hematopoietic cells involved, such as lymphoid, myeloid, mixed or undifferentiated, these disorders are classified accordingly. In contrast, chronic leukemias encompass a broad spectrum of diseases characterized by uncontrolled proliferation and expansion of mature, differentiated cells of the hematopoietic system. Thus, chronic leukemias are classified depending on the type of hemopoietic cells involved.

## 2. Age and Race

Age and race are important factors in the incidence of leukemias. For example, in the United Kingdom, 42.8% of all leukemias occur in individuals over 65 years of age [2]. A review of the subject in the United States reports that the overall age-adjusted leukemia incidence is highest in the White population at 15 per 100,000, followed by Blacks at 11 per 100,000, and Hispanics at 10.6 per 100,000 population [3]. The incidence among Asian/Pacific Islanders was 7.8 per 100,000 and in American Indian/Alaskan Natives, 8.3 per 100,000 population [3]. Similar racial and ethnic patterns were found for age-adjusted mortality rates per 100,000 population, which were 7 for Whites, 5.6 for Blacks, 4.8 for Hispanics, 3.8 for Asian/Pacific Islanders and 3.3 for Indian/Alaskan Natives [3].

While leukemia affects all age groups, its distribution varies based on the type of the disease. The age-adjusted incidence rate of leukemia from 2012 to 2016 in the United States in children, adolescents and young adults younger than 20 years was estimated to be approximately 4.6 per 100,000. Approximately 4.8 percent of all leukemia and lymphoma cases were diagnosed in individuals younger than 20 years of age. As such, it constituted approximately 20–30% of all cancers in this age group. Acute lymphoblastic leukemia (ALL), which is most common in childhood and adolescence, accounts for approximately 75% of all leukemia cases in individuals under 20 years of age and approximately one-quarter of all pediatric cancers. The peak incidence is in children ages 2–5 years. On the contrary, acute myeloblastic leukemia (AML), with an overall incidence of 3–5 cases per 100,000 in the general population, is far more prevalent in adults, with an incidence of only 7.7 per million between the ages of 0–14 years. Indeed, the median age at diagnosis for AML is 66 years with 54% of patients diagnosed after age 65 and 33% over age 75. It is of note that the incidence of acute lymphoblastic leukemia has increased over time, at least in the pediatric age group. This, in part, has been attributed to the improved accuracy of the diagnostic techniques and reporting. The incidence for AML appears to have been unchanged [4].

As noted above, racial differences in the incidence of and mortality caused by different types of acute leukemias have been reported. The incidence for some leukemias, such as ALL, is higher in the White population than the Black population. Based on a 2011 report by the American Association for Cancer Research, Hispanic children have the highest incidence of ALL and one of the lowest survival rates among pediatric patients diagnosed in the United States. In this regard, the report indicates that the overall mortality risk for ALL is 45 and 46 percent greater for Blacks and Hispanics than the White population, respectively. The rate of increase death in AML was 12 and 6 percent higher in these two groups than in the White population [5].

Despite reported differences in race and ethnicity in the incidence and outcome, the underlying causes for this disparities remain poorly understood [6,7,8,9,10]. While the role of the financial and social structure versus genetic factors in the incidence and outcome in various populations have been considered, the underlying causes are poorly recognized. Advances in genomic profiling, including genome-wide gene expression, DNA copy number, and single nucleotide polymorphism (SNP) genotype, may shed some light regarding the roles of genetics in these disparities [11].

## 3. Genetics

Undoubtedly, genetics plays a major role in the etiology of leukemia. A vast body of new literature regarding the relation of genetic factors in normal hemopoiesis and transition to acute leukemias and mechanisms of leukemogenesis is available; however, a summary of this is well beyond the scope of this general review. The effects of genetics in the pathogenesis of acute leukemias are most evident in identical twins. If one of the identical twins develops the disease before the age of 7 years, the other has twice as much chance of developing this disease than the general population. The chance of developing leukemia then reduces in time. The twin who reaches age 15 years without developing leukemia appears not to have a higher risk of developing this disease than the average population [12,13].

While for the majority of leukemia cases there are no obvious known predisposing factors, some genetic and acquired germline mutations and clonal chromosomal abnormalities are associated with increased incidence of leukemia [14]. Increasingly, using genome-wide association studies, germline mutations that can cause leukemia prone changes have been identified. Patients with DNA repair disorders and constitutional chromosomal anomalies can be predisposed to the development of leukemia. Some inherited mutations have the potential to enhance the risk of developing leukemia, but this is in the absence of extramedullary phenotypes. Some families have an increased incidence of leukemia with no known inherited mutations [14]. The major inherited and genetic disorders resulting in a predisposition to acute leukemia are summarized in Table 1. The identified genes that can be inherited in an autosomal dominant fashion and potentially result in the development of leukemia include *CEPBA*, *RUNX1* and *GATA2*. The *CEBPA* gene, located at chromosome19q13.1, encodes granulocytic differentiation factor *C/EBPa*, a member of the *bZIP* family of proteins. The *RUNX1* gene is located at 21q22.12 and is a transcription factor involved in hemopoiesis. The *GATA2* gene is located at 3q21.3 and is involved in preserving the integrity of hematopoietic stem cells and regulating phagocytosis. Mutation of *GATA2* has been associated with congenital neutropenia and MonoMAC syndrome, a disorder that frequently results in myelodysplastic syndrome (MDS), increased rate of infections and AML or chronic myelomonocytic leukemia [14]. Monosomy 7 has been reported in families with multiple members having MDS and AML. Likewise, inherited bone marrow failure syndromes with a variety of genetic abnormalities can lead to the development of leukemia [14].

DNA damage and impaired DNA damage repair capacity during proliferative DNA replication can result in mutations leading to the initiation of malignant transformation. Clonal hematopoiesis of indeterminate potential (CHIP) is a common asymptomatic condition that increases the risk of the development of hematological malignancies by the expansion of age-related somatic mutations in the hematopoietic lineages of aging individuals. CHIP-associated mutations and genetic alterations in the hematopoietic stem cells and progenitors have the potential to redirect their development and result in pathogenesis of hematological malignancies. Recent progress in the study of cancer genomics and single-cell molecular analysis has made it possible to study the clonal population and their genetic and time-related sequences of genetic and epigenetic evolutions in detail. These may help to better understand the role of clonal evolution in lymphoid and myeloid leukemia as a principal driver in the disease initiation, progression and resistance. Genome-wide analysis has resulted in better understanding of various signaling pathways and their role in the development of leukemias. Further genomic and epidemiologic research will likely lead to the identification of new constitutional genetic mutations and their relation to the environmental mutagens and pathogens in order to initiate malignant hematological clones and progress to full-fledged leukemia [15,16].

Events associated with leukemic transformation are often a multifactorial process. For example, the *ETV6-RUNX1* fusion gene, which is found in approximately 25% of ALL cases in the pediatric age group, is acquired in utero; however, it needs a secondary somatic mutation to be activated. In one study, using exome and low-coverage whole-genome sequencing to evaluate the events culminating in this oncogenic rearrangement, *RAG*-mediated deletions emerged as the prominent driving force in this mutational process. Combining the data of point mutations and rearrangements identified *ATF7IP* and *MGA* as tumor-suppressor genes in ALL. Therefore, a multifactorial mutational process, targeting the promoters, enhancers and first exons of genes, which normally regulate B-cell differentiation, transforms *ETV6-RUNX1*-positive lymphoblasts [17,18].

Individuals with certain genetic disorders are known to have an increased rate of developing leukemia [14,15,16,17,18,19,20,21,22,23,24,25,26,27]. Persons with disorders such as Li Fraumeni syndrome, Down syndrome, Shwachman syndrome, neurofibromatosis type 1, Fanconi’s anemia, ataxia-telangiectasia, Bloom syndrome, Klinefelter syndrome and Diamond Blackfan anemia are at higher risk for the development of leukemia [19,20,21,22,23,24,25,26,27]. Point, missense or nonsense mutations can occur in the tumor suppressor genes, some of which encode for proteins with suppressive effects on the regulation of the cell cycle and ability to promote apoptosis. Disorders such as Noonan syndrome and CBL syndrome are associated with a high risk of leukemia [14]. Bone marrow failure syndromes such as thrombocytopenia with absent radius, amegakaryocytic thrombocytopenia, dyskeratosis congenita and severe congenital neutropenia, including Kostmann syndrome, are also associated with a higher risk of the development of leukemia [14]. Myelodysplastic syndrome, myelodysplasia, polycythemia vera and primary thrombocythemia are also found to be associated with the increased rate of this disease [14]. DNA repair defects, such as mismatch repair deficiency syndrome, which involves sporadic mutations in genes responsible for DNA repair, as seen in some variants of Lynch syndrome, ataxia-telangiectasia, Nijmegen breakage syndrome, Bloom syndrome and Fanconi anemia, can be associated with hematological malignancies [14,19]. It is of note that approximately 33% of patients with Fanconi anemia develop a hematological malignancy by age 40. Germline polymorphisms in *IKZF1, CDKN2A, CEPBE* and *ARID5B* have been shown to be associated with increased risk of ALL [14].

Patients with various primary inherited immunodeficiency syndromes are predisposed to the development of malignant disorders, including leukemia [14]. These include Wiskott–Aldrich syndrome (an X-linked disorder with the triad of thrombocytopenia, immune deficiency and eczema) and Bruton agammaglobulinemia, which is due to the Bruton tyrosine kinase gene located at the chromosome Xq21.3-22 location.

As discussed in the AML and ALL sections of this review, two-hit theories involving genetics for the development of leukemias have been proposed.

## 4. Environment and Occupations

A large number of environmental causes for the development of leukemia have been suggested. These mostly involve exposure to cancer-causing agents, including chemicals, infections and radiation during various stages of life [28,29,30,31,32,33,34,35,36,37]. Certain exposures, occupations, industrial hazards, and hobbies have been implicated in a higher risk of leukemia [34,35,36,37]. There is a reported variation in the incidence of leukemia based on the type of industry (Table 2). The relation of certain occupations and the occurrence of acute leukemias is not certain and at times controversial. Occupations described to be associated with increased risk for leukemias include, but are not limited to, agricultural and forestry work and crop production [38,39,40,41,42,43,44,45] with exposure to pesticides and fertilizers [46,47,48], construction [33], animal slaughtering and poultry work [49,50], vocations in the oil/gas industries with exposure to benzene [40,48,51,52,53,54,55,56,57], oil refining and petrochemicals [58,59,60,61,62,63,64], automobile mechanic works [42,65], electrical utility careers, jobs with exposure to magnetic fields [66,67,68,69,70,71], works in the nuclear power industry/exposure to ionizing radiation [55,56,72,73,74,75,76], furniture manufacturing/repair [77] and nursing and health care positions with exposure to infectious agents/viruses [42,44,78,79,80]. Other occupations with increased risk of leukemia include hairdressing and hair dying [55,81], painting [82,83,84,85], laundry work, dry-cleaning with exposure to dry-cleaning chemicals [42,86,87], teachers [40,42], workers in the shoe and boot manufacturing industry [88] and taxi, bus, truck, and railway drivers and conductors [55,58,71,89,90]. Occupations with exposure to alkylating agents and formaldehyde [78,91,92], textile workers and manufacturers [55,93] and semiconductor workers [55,94] are also found to have a higher risk of leukemia. Increased risk of leukemia due to contact with workers in these industries is suggested [90]. With a few exceptions, such a diverse range of occupations, without a unifying element, lacks specificity (Table 3).

Direct and indirect exposure to chemicals and pesticides, in a variety of occupations, has been reported as a cause for the development of leukemia [95,96,97]. Likewise, exposure to hydrocarbon compounds, such as benzene, gasoline and trichloroethylene, in person or indirect, has been implicated in the development of leukemia [64,98]. Likewise, in children, a multiplicity of factors, including parental occupations, have been proposed to increase the risk of acute leukemia. A review of the subject is available [99].

## 5. Effects of Radiation

The effects of ionizing radiation in the development of leukemia at various phases of life, including preconception, in utero, and post-natal exposures, have been proposed, and various examples have been published. A correlation between the dose of irradiation and the occurrence of leukemia has been reported [100,101]. Following the bombing of Hiroshima and Nagasaki, Japan, the rate of occurrence of leukemia among survivors who were within 1000 meters of the explosions was 20-fold higher than the general population [101]. More recently, the consequences of the Chernobyl nuclear power plant accident have been studied [102,103,104].

There are conflicting data regarding the risk of the development of leukemia and exposure to diagnostic x-rays. Some studies have shown an elevated risk for childhood leukemia related to paternal diagnostic X-rays. An increased risk was found if two or more X-rays of the lower abdomen were done. However, no increased risk was noted if the data were restricted to lower abdominal X-rays [105,106]. In one study, no increased risk of leukemia was found with maternal abdominal X-rays. Some evidence of increased risk in offspring was noted if the father had more than one abdominal X-ray done before conception or had a prior intravenous pyelogram [103]. 

In general, a correlation between diagnostic X-rays and the development of leukemia is inconsistent, inconclusive and subject to a number of variables, including time and reason for the procedure and statistical errors. Some studies have reported an increased rate of leukemia in individuals who have had diagnostic X-rays. For example, in one report, children having one or more computed tomography scans had an increased ratio of developing leukemia, indicating that even with low doses of ionizing radiation, there is an increased risk of this disease. However, others find no correlations between diagnostic X-ray tests and increased risk, especially if tests done close to the time of diagnosis of leukemia are subtracted.

Inconsistent results regarding exposure to nonionizing radiation as an etiological factor for the development of leukemia have been reported, and generally, the effects of such exposures in the development of this disease have been disputed [107,108,109,110,111,112,113].

## 6. Prior Immunosuppressive and Chemotherapy

Individuals who have received chemotherapy for the treatment of cancer, with or without radiation, have an increased risk of leukemia. A variety of immunosuppressive therapies can also increase the risk of developing acute leukemias. Certain chemotherapy agents, such as alkylating agents, platinum derivates and topoisomerase II inhibitors, are associated with higher risk for the development of this disease. The addition of radiation therapy to chemotherapy increases the risk involved [114,115]. In a study of 82,700 women with invasive breast cancer, the risk of acute nonlymphocytic leukemia was significantly higher for all types of therapy, including regional radiotherapy alone, alkylating agents alone and the combination of chemotherapy and radiation. The relative risk was 2.4 for radiotherapy, 10 for alkylating agents and 17.4 for the combination. The observed risk for the development of leukemia was dose- and treatment-dependent, with melphalan having ten times more leukemogenic effect than cyclophosphamide. With total cyclophosphamide doses of less than 20,000 mg, only a small increase in the risk of secondary leukemia was observed.

## 7. Parental and Residential Factors

In the pediatric age group, paternal hobbies and occupations, such as work involving contact with gasoline, paint, pigments, solvents, pesticide and plastics, jobs in metal, textile, pharmaceutical industries and professions requiring engine repair, have been investigated for the development of leukemia in children. Direct and indirect effects of chemical agents on children, including via breastfeeding and exposure to contaminated clothing or environment, have been implicated [64,95,96,97,110,116,117,118,119,120,121,122]. Likewise, household exposure to pesticides and insecticides has been found to be associated with a higher risk of leukemia in the pediatric age group [123].

The proximity of place of birth to the industrial sites with release of volatile organic compounds has been reported [124].

Children born through in vitro fertilization have a higher rate of acute leukemia and Hodgkin’s lymphoma [125,126]. 

Parental alcohol consumption and smoking during prenatal and neonatal periods and childhood have been suggested to contribute to the development of leukemia in their offspring. The risk may be related to the severity, frequency, duration and extent of the exposure [127,128,129,130,131,132]. Maternal use of marijuana during and after pregnancy has been reported to increase the risk of ALL and AML by 10-fold [133]. Effects of the chemico-biological interactions have been explored.

Several studies regarding the etiology of leukemia in childhood have examined the relation of the parent’s age, maternal history of fetal loss, birth characteristics and higher birth weight. A positive trend associating maternal and paternal age greater than 35 and 40 years, respectively, and the occurrence of ALL in the offspring was found. Maternal and paternal age exceeding 40 years has been reported to be associated with an increased the odd ratios of 1.95 and 1.45 for the development of childhood leukemia [134]. Maternal history of fetal loss and the risk of the development of ALL and AML is conflicting. This may reflect genetic predisposition or the effects of the environment [135,136,137]. Increased rate with the higher birth weight is presumed to reflect an increased ability for cell proliferation [138].

## 8. Infections

Infections, including bacterial, viral and fungal agents alone, and in conjunction with genetic mutations, have been implicated in leukemogenesis. Infective agents have been suspected to be associated with the development of cancer in general and acute leukemias in particular [139,140,141,142]. However, save for some recent reports [141,142], no consistent agent which can be uniformly applied to a group of patients is available. The impact of a variety of infectious organisms, including the Epstein–Barr virus (EBV), herpesvirus, human immunodeficiency virus (HIV), severe acute respiratory syndrome (SARS), COVID-19 and Human T-lymphotropic virus (HTLV-1), among others, in the development of leukemia have been hypothesized and explored [99,139,143,144,145,146,147,148,149,150,151,152,153].

Associations, such as that of EBV and Burkitt’s lymphoma in the endemic area of Eastern Africa, are well recognized. While such an association is strong, the findings are not entirely applicable to all cases; for example, the 8;14 chromosomal translocation resulting in constitutive activation of c-Myc oncogene, p53 mutations, variation in viral gene expression in some patients, actions of EBV oncoproteins and many other similar factors complicate such an association. During the first two years of life, exposure to EBV resulting in a positive serological response and the development of Burkitt’s lymphoma has been reported [139,149,154].

Human T-cell leukemia virus type 1, which is also known as human T-lymphotropic virus (HTLV-1) has been linked to adult T-cell leukemia/lymphoma (ATL), presumably due to the insertion of their DNA or RNA into the host cell. HTLV-I is proposed to cause ATL, after a latent period, in approximately 5% of carriers. It is hypothesized that post infection, HTLV-I advances the in vivo clonal proliferation of HTLV-I infected cells through the functions of encoded viral proteins, including Tax [155]. If exposed to pathogens, mice with monoallelic loss of the B-cell transcription factor *PAX5* are predisposed to B-cell ALL [147,156].

Carcinogenic effects of fungal agents and aflatoxin are well established, but the mechanisms resulting in this phenomenon are not entirely clear. Few reports of fungal isolation from residences of patients with leukemia, including ALL, are available [157,158,159,160], and generally, their carcinogenic impacts are attributed to immunosuppression [158,159]. Mycotoxin-producing fungi, isolated from a residence associated with four patients with leukemia from three families has been reported and the leukemogenesis attributed to the mycotoxins’ immune depressive effects [158]. A published report is available regarding the isolation of fungal agents from a house where a husband had acute myelomonocytic and the wife had undifferentiated leukemia. The report indicates that the extract of the fungal isolates resulted in the depression of the phytohemagglutinin skin test in guinea pigs while the control was negative [159]. In another report, using the supernatant of the culture of *Aspergillus* and a modified microimmunodiffusion technique, sera from 36 patients with cancer, 15 of whom had leukemia or lymphoid tumors, positive results were found in 30% of the participants with cancer as compared to only 6% of the controls. This effect was attributed to a reaction to the aflatoxin produced by the fungi [158].

In recent reports, an *Aspergillus Flavus* species, isolated from the home of a patient with ALL by electron microscopy examination was found to contain a mycovirus within the body of the fungi and its culture supernatant [141,142]. This organism, by chemical analysis, was reported not to produce any aflatoxin. Lack of production of aflatoxin, of course, is well known to occur in fungal organisms that host a virus, providing a stage for virus–virus and virus–host interactions, which results in blocking such production [161,162,163]. This study reports that using an ELISA technique, the plasma of patients with ALL in a complete remission, and long-term survivors, had antibodies to the products of this mycovirus containing *Aspergillus Flavus*. This test was reported to be able to recognize and differentiate patients with ALL in remission from those of normal controls, as well as individuals with sickle cell disease and solid tumors [141]. In a related study, exposure of the peripheral blood mononuclear cells (PMBC) from individuals with ALL in a complete remission to the products of the culture of the above mycovirus containing *Aspergillus flavus* had resulted in the reproduction of genetic markers and cell surface phenotypes characteristic of ALL. Controls were found to be negative. A serial timeline evaluation to examine the time required for the development of ALL cell surface phenotypes, using flow cytometry, had revealed that the conversion from normal to leukemic cell surface markers had started shortly after incubation with the supernatant of the mycovirus containing *Aspergillus Flavus* and was completed over a 24-h period. The report indicates that the addition of EBV to the mixture had not altered the results. In these experiments, aflatoxin which was used as a positive control, indiscriminately had produced abnormal cell surface phenotypes in the PBMCs from normal controls as well as the ALL patients in a full remission. This study may indicate that mycovirus containing *Aspergillus Flavus* can directly affect cells of ALL patients in remission and alter and transform the genetic and cell surface makers of their presumably genetically susceptible cells, and not controls. The report also indicates that in limited studies, when cultures with and without EBV were irradiated, this had significantly increased the co-expression of CD10/CD19, which is considered as one of the characteristic cell surface phenotypes in the ALL [142]. Considering the two-hit theory for the development of acute lymphoblastic leukemia, it is postulated that the mycovirus containing *Aspergillus Flavus* may provide a consistent organism in the mechanism of leukemogenesis in acute lymphoblastic leukemia [141]. These experiments may give credence to the idea that a combination of pre-existing genetics/epigenetics background and exposure to infections in the environment may result in the development of ALL [142].

Mycotoxins produced by fungal agents, including aflatoxins, ochratoxin A, fumonisins, certain trichothecenes, and zearalenone, are known to be carcinogenic [164]. Some mycotoxins, such as Patulin and Gliotoxin, have a toxic epipolythiodioxopiperazine metabolite with substantial immunosuppressive effects. These agents can cause apoptosis in PBMC and have selective in vitro cytotoxicity, as compared to others that have suppressive effects on the immune response [165,166]. Gliotoxin, in vivo, is reported to inhibit the transcription of NF-ϰB in response to a number of stimuli in T and B cells. It is reported that in high concentrations, this agent is able to prevent the binding of NF-ϰB to DNA in vitro [167]. The presence of NF-ϰB p65 (Rel A) is required for protection from TNA-α. It is of note that constitutively activated NF-ϰB complexes have been reported in the majority (39/42) of the patients with ALL without any subtype restrictions [168].

The correlation between exposure to infections, including fungal organisms, and occupations with increased rate of leukemia, such as agricultural work, which potentially exposes the workers to fungal and other agents is not clear and requires future investigation.

As noted before, in recent years, two-hit hypotheses, indicating multifactorial causation for the development of acute leukemias, have been proposed [169,170]. The specifics for two major acute forms of the disease, i.e., acute lymphoblastic and myeloblastic leukemia, are described in the following sections.

## 9. Acute Myeloblastic Leukemia

Acute myeloblastic leukemia (AML) has two peaks in occurrence, during early childhood and later in adults. The median age for the newly diagnosed patients with AML is 66 years. While the disease can occur at any age, the diagnosis is relatively rare before 40 years of age. Based on the United States statistics obtained from 2000 to 2004, the incidence in individuals under age 65 is 1.7 per 100,000, with the rate increasing to 16.8 per 100,000 in those aged 65 or older. The incidence of AML varies with gender and race. Overall, in the United States Surveillance Epidemiology and End Results Program (SEER) database for children aged 1–4 years, the recorded incidence rate is 0.9 per 100,000 for boys and 0.8 for girls [171].

During the first few years of life, the incidence of AML in Whites is three-fold higher than in blacks; however, black children have slightly higher rates of this disease after this age. In the United States between 2000 and 2004, with the rate of 3.7 per 100,000 population, AML was more common in Whites than Blacks who had a rate of 3.2 per 100,000 [171]. The increased incidence with age is partially suspected to be due to the progression of myelodysplastic syndromes (MDS) to AML. The MDS-related AML is characterized by common cytogenetic abnormalities, which is shared with MDS and has an increased frequency of unfavorable prognosis. In children, the incidence of acute myeloblastic leukemia between 2005 and 2009 was estimated to be 7.7 cases per million for ages 0–14 years. In the pediatric age group, the peak incidence rate occurs in the first year of life followed by a steady decline up to the age of 4 years. In infants less than one year of age, the incidence is 18.4 per million [2,172]. The incidence of AML has remained relatively constant in children and adults, with the exception of a slight increase in the oldest age group [173].

Other than MDS, in most cases, the etiology of AML is unclear. A significant amount of information and knowledge concerning leukemogenic agents, especially chemotherapy regimens used for the treatment of a variety of malignant disorders, has accumulated [114,115]. Associations of certain molecular pathogenesis such as t(8;21) translocation and inversion of chromosome 16 in AML have been reported. In addition to these genetic alterations, epigenetic changes, such as promoter silencing by hypermethylation of the p15/INK4b and other genes in the pathogenesis of AML, have been recognized. The association of certain genetic factors, including genetic defects, and AML, especially in children, is suggested. As noted above, patients with a variety of genetic disorders, such as Down’s syndrome, have a substantially higher potential for the development of malignant disorders, including AML. For example, children with Trisomy 21 have a 10- to 20-fold increased potential of developing acute leukemia, mostly AML [21,26,174,175].

Acquired genetic and clonal chromosomal abnormalities are found in 50–80% of AML patients, especially in older individuals and those with secondary leukemia. These abnormalities include loss or deletion of chromosomes 5, 7, Y and 9. Chromosome translocations, including those of t(8;21) (q22;q22), t(15;17) (q22;q11), trisomy 8 and 21, and abnormalities in the chromosomes 16, 9, and 11, have been reported [2,176,177,178,179,180,181,182,183,184]. Cases of tetraploid acute leukemia have been recorded. In one report, a pseudodiploid clone characterized by t(8;21) and a hypotetraploid clone with two t(8;21) and a loss of two Y chromosomes was recorded [185]. A report of specific associations between the most frequent balanced translocations in AML, such as AML with the (8;21) translocation and inversion of chromosome 16, and acute promyelocytic leukemia with the (15;17) translocation, is available. Regarding the pathogenesis of AML, in addition to these genetic alterations, epigenetic events such as promoter silencing by hypermethylation of the p15/INK4b and other genes have been reported.

A “two-hit-hypothesis” for the development of AML phenotype by class I and II mutations has been proposed. This two-hit hypothesis is different from that proposed for ALL. Among candidates for class I are mutations in *FLT3, N-RAS* or *K-RAS*. Class II mutations are exemplified by *RUNX1-RUNX1T1* (also known as *AML1-ETO*, *RUNX1-MTG8*), *CBF/SMMHC*, *PML/RAR*, and MLL-related fusion genes. An example for this hypothesis is activating mutations in *FLT3*, which is seen in all subtypes of AML and can confer a proliferative advantage to the hematopoietic progenitors, (class I) and gene rearrangements affecting one of the hematopoietic transcription factors (class II). A combination of class I and class II mutations are necessary for the proposed theory to result in the development of AML [169,186]. This theory is in line with an increased rate in the development of AML in individuals treated for other malignant disorders [114,115].

Risk factors for the development of AML, as outlined before, include exposure to radiation, chemicals and engagement in various occupations and hobbies. 

In some forms of acute promyelocytic leukemia (APML), distinct chromosomal and gene-rearrangement aberrations have been recognized. These may be different in various areas of the world. For example, while increased incidence of APML in adult patients originating from Latin America and in children in Southern Europe has been reported, the genetic rearrangement in these two localities is different. This may indicate that a particular breakpoint site may be responsible in various locations. It is known that certain polymorphisms in the genes metabolizing carcinogens are associated with an increased risk of AML. For example, NAD(P)H:quinone oxidoreductase 1 (NQO1) is a carcinogen-metabolizing enzyme that detoxifies quinones and reduces oxidative stress. A polymorphism at nucleotide 609 of the NQO1 complementary DNA decreases the activity of these enzymes and can result in therapy-related AML [169,187,188].

## 10. Acute Lymphoblastic Leukemia

Acute lymphoblastic leukemia (ALL) is the most frequently diagnosed cancer in the pediatric age group, amounting to approximately 25–30% of all childhood malignant disorders. The annual incidence of acute lymphoblastic leukemia in the United States is approximately 4.6 cases per 100,000 between the ages 0–14 years, with a peak incidence at age 2–5 years. The incidence of ALL during the first year of life is slightly higher in females than in males [189].

Similar to other leukemias, the role and possible effects of a number of factors, as outlined above, for the development of ALL have been proposed. The effects of various environmental factors, including parental preconception, in utero and post-natal exposure to ionizing radiation, have been explored. Likewise, the risks of nonionizing radiation, chemicals, infections, hydrocarbons and pesticides have been evaluated. The effects of parental alcohol, cigarette, and illicit drug use in the development of ALL in offspring have been examined.

Genetics play a major role in the development of leukemia in general and acute lymphoblastic leukemia in particular. The importance of genetics is most evident based on the concordance studies on identical twins with leukemia [12,190,191].

The concept that some cases of leukemia originate in utero by leukemogenic translocations or clonotypic gene fusion sequences is intriguing. Siblings of children with leukemia have a higher risk of developing this disease than others, albeit a relatively minimal risk [192,193].

It is well recognized that some genetic disorders, including Down syndrome [20,21] Shwachman syndrome [22], neurofibromatosis [23], Fanconi anemia [24], Bloom syndrome [25,26], and ataxia-telangiectasia [27], are associated with the increased rate of leukemia. Some of these syndromes, such as Down and Bloom syndromes and Fanconi anemia, have a higher incidence of AML than ALL. While genetic syndromes resulting in the development of ALL only accounts for a very small portion of the cases, the fact that they are associated with the increased rate of this disease points to the importance of genetics in the process of leukemogenesis. In B-cell ALL, genetic alterations, which are specific to each ALL immunophenotype, include hyperdiploidy, hypodiploidy, *BCR-ABL1, ETV6-RUNX1* or *TCF3-PBX1* fusions, *PAX5* or *ETV6* mutations, MLL rearrangements, or intrachromosomal amplification of chromosome 21 (iAMP21) specific for B-ALL. Alterations in *LMO2*, *TAL1*, *TAL2*, *TLX1*, *TLX2*, or *HOXA* are characteristics of T-cell ALL [186,194].

A revised taxonomy of B-ALL highlights the genetic heterogeneity of this disease by incorporating 23 subtypes, defined by chromosomal rearrangements, sequence mutations or heterogeneous genomic changes. Most of these molecular changes are acquired and not inherited [195]. Epigenetic priming in pediatric ALL has been suggested [196].

A recent two-hit theory combines genetic mutation and exposure to one or more infections for the genesis of ALL. The revised two-hit hypothesis for the development of precursor B-cell ALL hypothesizes that this disease arises through a two-step process. The first step is a predisposing genetic mutation. The second step suggests exposure to one or more infections [170]. This proposal, therefore, hypothesizes that the process of developing ALL begins in utero by fusion gene formation or hyperdiploidy and preparation of pre-leukemic clone. It is estimated that step one of the process occurs in approximately 5% of newborns; however, only one percent of those that are predisposed progress to develop ALL. The hypothesis suggests that exposure to infections during early life can protect the individual from the development of precursor B-cell ALL. In the absence of early exposure, in a small fraction of the population, exposure to infection later in life triggers the critical secondary cellular mutations. 

In western industrialized countries, approximately 80% of the cases of B-lineage ALL have either an *ETV6/RUNX1* translocation or a high-hyperdiploid leukemic clone. These are proposed to have been initiated in utero. Only one percent of healthy newborns have translocation t(12;21)[*ETV6/RUNX1*]-positive cord blood cells. In the developed countries, a lower chance of exposure to infections in early life is proposed to be the reason for a relatively higher rate of ALL in children. In contrast, in developing countries, a higher rate of exposure to infections, and possibly malnutrition, is suggested to contribute to a reduced rate of childhood ALL. These factors are proposed to increase the cortisol secretion during infections and the cellular response to cortisol [197].

Although the sequence of the events cannot be ascertained with any precision, and a number of alternatives exist, genetic predisposition along with random exposure to an infective agent can be plausible. While several genetic mutations have been suggested, no infection category or specific agent has been proposed. Recent reports suggest a mycovirus containing *Aspergillus flavus* as one of the possible candidates for the infection category [141,142].

## 11. Conclusions

Acute leukemias account for a significant portion of malignant disorders. These malignancies occur universally, albeit with different rates in various areas of the world, and affect all age groups, including children. While association of significant causative factors for the development of acute leukemias has been reported, the etiology of these disorders remains unclear. Recent advances in genetic and epigenetics provide indications for their involvement in leukemogenesis in acute leukemias. Likewise, the effects of environmental factors, including infections, have been explored. A recent finding of an antibody to a mycovirus containing *Aspergillus* flavus in patients with ALL in full remission and re-development of genetic and cell surface phenotypes, characteristic of ALL, upon exposure of PBMN cells from these patients, and not normal controls, to the products of this organism, may provide a new venue for research in leukemogenesis. More research to fulfill the required tenants of theories regarding the development of acute leukemias based on the combination of genetics and environment is needed.

## Figures and Tables

**Table 1 cancers-13-02256-t001:** Inherited predisposing syndromes to hematologic malignancy.

Predisposing Disorder	Gene	Inheritance	Type of Leukemia
CEBPA	*CEBPA*	AD	MDS/AML
Monosomy 7	*7p/q*	AD	MDS/AML/ALL
Familial platelet disorder/AML	*RUNX1*	AD	MDS/AML/T-cell ALL
MonoMAC Syndrome	*GATA2*	AD	MDS/AML
Familial AML with mutated DDX41	*DDX41*	AD	MDS/AML/CMML
Thrombocytopenia 2	*ANKRD26*	AD	MDS/AML
Thrombocytopenia 5	*ETV6*	AD	MDS/AM/CMML, B-cell ALL
Familial MDS/AML with mutated GATA2	*GATA2*	AD	MDS/AML/CMML
Li-Fraumeni syndrome	*TP53*	AD	ALL
Neurofibromatosis type 1	*NF1*	AD	JMML/MDS/AML
Noonan syndrome	*PTPN11*	AD	JMML/MDS/AML
CBL syndrome	*CBL*	AD	JMML
Familial aplastic anemia with mutated SRP72	*SRP72*	AD	MDS/AML
Familial B- cell ALL with mutated PAX5	*PAX5*	AD	ALL
Germline SH2B3	*SH2B3*	AR	ALL
Telomere syndromes (dyskeratosis congenita)	*TERC*, *TERT*, *CTC1*, *DKC1*, *NHP2*, *NOP10*, *RTEL1*, *TINF2*, *WRAP53*, *ACD*, *PARN*	AD, AR	MDS/AML
Diamond Blackfan anemia	*RPS19*, *RPL5*, *RPL11*	Sporadic, AD, AR,	MDS/AML/ALL
Shwachman–Diamond syndrome	*SBDS*	AR	MDS/AML/ALL
Amegakaryocytic thrombocytopenia	*c-MPL*	AR	MDS/AML
Thrombocytopenia with absent radii syndrome	*RBM8A*	AR, Sporadic	ALL/AML
Severe congenital neutropenia	*ELA2*, *HAX1*, *G6PC3*, *WASP*	AD, AR, X-linked	MDS/AML
Fanconi anemia	*FANCA*, *FANCB*, *FANCC*, *BRCA2*, *FANCD2*, *FANCE*, *FANCF*, *FANCG*, *FANCI*, *BRIP1*, *FANCL*, *FANCM*, *PALB2*, *RAD51C*, *SLX4*	AR	ALL/AML
Mismatch repair Cancer syndrome	*PMS2*, *MSH6*, *MLH1*, *MSH2*	AR	ALL
Ataxia-telangiectasia	*ATM*	AR	ALL
Nijmegen breakage syndrome	*NBS1*	AR	ALL
Bloom Syndrome	*BLM*	AR	ALL
Werner Syndrome	*WRN (RECQL2)*	AR	MDS/AML
Rothmund–Thomson	*RECQL4*	AR	AML
Wiskott–Aldrich Syndrome	*WASP*	X-linked	ALL
Burton’s agammaglobulinemia	*BTK*	X-linked	ALL
Trisomy 21 (Down Syndrome)	*21q*	Sporadic	ALL/AML

AD—autosomal dominant, AR—autosomal recessive, MDS—myelodysplastic syndrome, ALL—acute lymphoblastic leukemia, AML—acute myeloblastic leukemia, JMML—juvenile myelo-monocytic leukemia, CMML—chronic myelomonocytic leukemia.

**Table 2 cancers-13-02256-t002:** Industries with Increased and Decreased Rate of Leukemia.

**Industries with Increased Rate of Leukemia**
Agriculture/Crop production and related ventures
Forestry
Fishing and Hunting
Construction and related services
Animal slaughtering/poultry processing
Oil refining and petrochemicals
**Industries with Decreased Rate of Acute Leukemia**
Professional, legal and technical services
Computer systems and related services
Business support, management and administrative services
Public administration

**Table 3 cancers-13-02256-t003:** Occupations with an Increased and Decreased Risk of Acute Leukemia.

**Occupations Associated with Increased Risk of Acute Leukemia**
Farmers, foresters, agriculture workers and related occupations
Fishing and related works
Construction, painting, maintenance and related occupations
Carpet, tile and floor installers
Building and ground cleaning, janitorial and maintenance workers
Healthcare workers
Workers exposed to solvents, chemicals and benzene
Electricians/electrical utility workers
Workers exposed to high doses of radiation/nuclear power industry
Automobile mechanics/drivers/rail conductors and pilots
Furniture manufacturers and repair personnel
Laundry workers, dry cleaners
Textile workers and manufacturers
Hairdressers
Teachers
**Occupations Associated with Decreased Risk of Acute Leukemia**
Attorneys and legal workers
Movers

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
