# Peer review of "Etiology of Acute Leukemia: A Review"

_cancers, 2021, doi:10.3390/cancers13092256_

Round 1
Reviewer 1 Report
This review does not discuss crucial issues about Acute Leukemia etiology in the genomic era.
First of all, there is no discussion of what is leukemia, what the differences are between a chronic and an acute leukemia, and of what are the clinical and biological features that are relevant in the development of the disease.
The WHO classsification is now led by mutational status, that is the key element in risk stratification and disease managing. In 2021 a review cannot ignore this approach.
That said, there is no adequate mentioning of how the new data on clonal hematopoiesis are providing new lights on the steps from normal to neoplastic hematopoiesis.
Finally, No hazard ratio data are provided on occupational/industrial/chemotherapy-related risks so that one cannot understand the relative risks.
Author Response
Response to reviewer 1:
Reviewer’s Question 1:"First of all, there is no discussion of what is leukemia, what the differences are between a chronic and an acute leukemia, and of what are the clinical and biological features that are relevant in the development of the disease. The WHO classsification is now led by mutational status, that is the key element in risk stratification and disease managing. In 2021 a review cannot ignore this approach".
Author’s Response : While the author fully agrees with the reviewer, please note the article is on the “Etiology of acute leukemias”. Therefore, the definition of leukemias, differences of acute and chronic leukemias and WHO classification is not within the scope of the article.
Reviewer’s Question 2: "That said, there is no adequate mentioning of how the new data on clonal hematopoiesis are providing new lights on the steps from normal to neoplastic hematopoiesis".
Author’s Response : Two paragraphs reviewing clonal hematopoiesis are added to the section 3 of the article.
Reviewer’s Question 3: Finally, No hazard ratio data are provided on occupational/industrial/chemotherapy-related risks so that one cannot understand the relative risks.
Author’s Response: An extensive review of the literature was again performed. No reliable data on the hazard ratio of various occupations regarding acute leukemias are available.

Reviewer 2 Report
Tebbi provides an excellent review of the known and hypothesized causes of acute leukemia. This is an excellent topic for the both pediatric and adult oncologists, as well as patients and families.
I found the writing to be excellent: easy to read, well sectioned, and good discussion on a number of points. Resources seem very comprehensive.
Critiques:
I would recommend adding that children/patients conceived by in vitro fertilization may have an increased risk of acute leukemia.
I would recommend expanding the section of world-regional geographic differences in ALL. [page 12, second to last full paragraph]. This is an important issue and requires more attention. ALL rates are significantly higher in developed countries rather than less developed regions. The authors touch on the "clean hypothesis," though would recommend that this be expanded more.
In the occupations paragraphs, I would add a line or two that there have been many negative studies of the listed occupations. Would clarify that most of the listed occupations are not considered to be significant risk factors for acute leukemia.
Author Response
Response to reviewer 2:
Reviewer’s Question 1: “I would recommend adding that children/patients conceived by in vitro fertilization may have an increased risk of acute leukemia”.
Author’s Response : A sentence regarding in vitro fertilization is added to the section 7 of the article
Reviewer’s Question 2: “I would recommend expanding the section of world-regional geographic differences in ALL”.
Author’s Response : The background section now includes regional differences.
Reviewer’s Question 3: “In the occupations paragraphs, I would add a line or two that there have been many negative studies of the listed occupations. Would clarify that most of the listed occupations are not considered to be significant risk factors for acute leukemia.”
Author’s Response: A sentence clearly stating that the data regarding the effects of occupations on the development of acute leukemia are controversial and uncertain is added.
Reviewer 3 Report
The current review article on the etiology of acute leukemia is written very well crafted and written. The review described several important factors and summarized very recent studies that described the etiology of hematological disorders. The review needs to correct following few minor points in the final version:
1) Check the first sentence beginning in '3.Genetics' section. Correct 'genetic' to 'genetics' or 'genetic makeup/structure' plays important role...
2) Correct "...to the development leukemia [14]." to "...to the development of leukemia [14]."
3) Table-3 was not cited the review. Please cite it appropriately in section '4.
4) Table-2 was cited wrongly in section 4 for the occupations (i.e. table-3). Cite table-2 correctly and elaborate in the text appropriately.
5) Remove 'b' letter before 'Animal salughtering/poultry processing' in Table-2.
6) Expand SARS and correct 'COVD-19' to 'COVID-19' in the first paragraph of section 8.
7) Indicate the abbreviation 'EBV' next to '.. including Epstein-Barr virus, herpesvirus..' as author used EBV in other next paragraphs of section 8.
8) Italicize 'Aspergillus Flavus' on page 9 (In recent reports,....paragraph last second sentence).
9) Italicize both 'in vitro' and 'in vivo' words on page 10 first paragraph similar to entire review.
10) Correct '..prevent binding of NF-kB DNA in vitro [164].' to '..prevent binding of NF-kB TO DNA in vitro [164].' Also check the following sentence which seems like incomplete. Correct 'TNA-α' to 'TNF-α'. [page-10 first paragraph]
11) Expand 'SEER' database in first paragraph of Section 9.
12) The references 111-194 were numbered twice in 'References' section. correct them
Author Response
Response to reviewer 3:
Reviewer’s Questions:
1) Check the first sentence beginning in '3.Genetics' section. Correct 'genetic' to 'genetics' or 'genetic makeup/structure' plays important role... Author’s Response : Corrected
2) Correct "...to the development leukemia [14]." to "...to the development of leukemia [14]." Author’s Response : Corrected
3) Table-3 was not cited the review. Please cite it appropriately in section '4. Author’s Response : Corrected
4) Table-2 was cited wrongly in section 4 for the occupations (i.e. table-3). Cite table-2 correctly and elaborate in the text appropriately. Author’s Response : Corrected
5) Remove 'b' letter before 'Animal salughtering/poultry processing' in Table-2. Author’s Response : Corrected
6) Expand SARS and correct 'COVD-19' to 'COVID-19' in the first paragraph of section 8. Author’s Response : Corrected
7) Indicate the abbreviation 'EBV' next to '.. including Epstein-Barr virus, herpesvirus..' as author used EBV in other next paragraphs of section 8. Author’s Response : Corrected
8) Italicize 'Aspergillus Flavus' on page 9 (In recent reports,....paragraph last second sentence). Author’s Response : Italicized
9) Italicize both 'in vitro' and 'in vivo' words on page 10 first paragraph similar to entire review. Author’s Response : Italicized
10) Correct '..prevent binding of NF-kB DNA in vitro [164].' to '..prevent binding of NF-kB TO DNA in vitro [164].' Also check the following sentence which seems like incomplete. Correct 'TNA-α' to 'TNF-α'. [page-10 first paragraph] Author’s Response : Corrected
11) Expand 'SEER' database in first paragraph of Section 9. Author’s Response :Expanded
12) The references 111-194 were numbered twice in 'References' section. correct them Author’s Response : Corrected
Reviewer 4 Report
- Abstract: Please mention that both AML and ALL are being discussed in the review manuscript.
- Genes should always be written in italics, and the modern nomenclature should be used (e.g. RUNX1-RUNX1T1 instead of AML1-ETO).
- Does the section on clonal hematopoiesis focus only on AML or also on ALL?
- Statement on NHL and acute leukemias showing an increased incidence after IVF: please provide a reference.
- Typo: Burkett, p. 9.
Author Response
Thank you for reviewing the manuscript and your comments. Response to your review is as follows:
- The fact that both ALL and AML are reviewed are added to the abstract.
- The following paragraph is added to the background:
Acute leukemias are malignant clonal disorders of blood forming organs involving one or more cell-lines in the hematopoietic system. These disorders are marked by the diffuse replacement of bone marrow with abnormal immature and undifferentiated hematopoietic cells, reduced numbers of erythrocytes and platelets in the peripheral blood. Based on the origin of the abnormal hematopoietic cells involved, such as lymphoid, myeloid, mixed or undifferentiated, the disorders are classified accordingly. In contrast, chronic leukemias encompass a broad spectrum of disorders characterized by uncontrolled proliferation and expansion of mature, differentiated cells of the hematopoietic system. Thus, chronic leukemias are classified depending on the type of hemopoietic cells involved.
- Extensive search of literature was done and no hazard ratio data regarding the rate of acute leukemias based on occupation was found.
Round 2
Reviewer 1 Report
The author did not answer two of my three questions: 1) there is no discussion of what is leukemia, what the differences are between a chronic and an acute leukemia, and of what are the clinical and biological features that are relevant in the development of the disease. There cannot be a Review on leukemia etiology not dealing with these crucial issues. 2) No hazard ratio data are provided on occupational/industrial/chemotherapy-related risks so that one cannot understand the relative risks. Author’s Response: An extensive review of the literature was again performed. No reliable data on the hazard ratio of various occupations regarding acute leukemias are available. Risks related to previous chemotherapy are very well studied.Author Response
Thank you for your review and comments. In response to your points, the following have been added/changed:
- AML1-ETO has changed to RUNX1-RUNX1T1. Genetic names which in process of printing were made non-italic are now italic again.
- The article mainly addresses AML and ALL.
- Two references are added for the increased rate of acute leukemias and Hodgkin's lymphoma for those born through in vitro fertilization.
- Burkitt misspell is corrected.
This manuscript is a resubmission of an earlier submission. The following is a list of the peer review reports and author responses from that submission.